# Gene Therapy for Parkinson’s Disease Using Midbrain Developmental Genes to Regulate Dopaminergic Neuronal Maintenance

**DOI:** 10.3390/ijms252212369

**Published:** 2024-11-18

**Authors:** Jintae Kim, Mi-Yoon Chang

**Affiliations:** 1Channelopathy Research Center (CRC), Dongguk University College of Medicine, 32 Dongguk-ro, Goyang 10326, Republic of Korea; jtmed92@gmail.com; 2Department of Premedicine, College of Medicine, Hanyang University, FTC12, 222 Wangsimni-ro, Seoul 04763, Republic of Korea; 3Biomedical Research Institute, Hanyang University, Seoul 04763, Republic of Korea; 4Hanyang Institute of Bioscience and Biotechnology (HY-IBB), Hanyang University, Seoul 04763, Republic of Korea

**Keywords:** gene therapy, Parkinson’s disease, dopaminergic neuron, adeno-associated virus, midbrain developmental genes, Nurr1, preclinical study, clinical trial

## Abstract

Parkinson’s disease (PD) is the second most prevalent neurodegenerative disorder. It is characterized by the progressive loss of dopaminergic (DAnergic) neurons in the substantia nigra and decreased dopamine (DA) levels, which lead to both motor and non-motor symptoms. Conventional PD treatments aim to alleviate symptoms, but do not delay disease progression. PD gene therapy offers a promising approach to improving current treatments, with the potential to alleviate significant PD symptoms and cause fewer adverse effects than conventional therapies. DA replacement approaches and DA enzyme expression do not slow disease progression. However, DA replacement gene therapies, such as adeno-associated virus (AAV)–glutamic acid decarboxylase (GAD) and L-amino acid decarboxylase (AADC) gene therapies, which increase DA transmitter levels, have been demonstrated to be safe and efficient in early-phase clinical trials. Disease-modifying strategies, which aim to slow disease progression, appear to be potent. These include therapies targeting downstream pathways, neurotrophic factors, and midbrain DAnergic neuronal factors, all of which have shown potential in preclinical and clinical trials. These approaches focus on maintaining the integrity of DAnergic neurons, not just targeting the DA transmitter level itself. In particular, critical midbrain developmental and maintenance factors, such as Nurr1 and Foxa2, can interact synergistically with neighboring glia, in a paracrine mode of action, to protect DAnergic neurons against various toxic factors. Similar outcomes could be achieved by targeting both DAnergic neurons and glial cells with other candidate gene therapies, but in-depth research is needed. Neurotrophic factors, such as neurturin, the glial-cell-line-derived neurotrophic factor (GDNF), the brain-derived neurotrophic factor (BDNF), and the vascular endothelial growth factor (VEGF), are also being investigated for their potential to support DAnergic neuron survival. Additionally, gene therapies targeting key downstream pathways, such as the autophagy–lysosome pathway, mitochondrial function, and endoplasmic reticulum (ER) stress, offer promising avenues. Gene editing and delivery techniques continue to evolve, presenting new opportunities to develop effective gene therapies for PD.

## 1. Introduction

Parkinson’s disease (PD) is the second most common neurodegenerative disease, affecting more than 6 million patients worldwide [1]. The pathologic hallmark of PD is the progressive loss of dopaminergic (DAnergic) neurons in the substantia nigra and decreased dopamine (DA) levels. Since DA regulates movement and the brain’s response to rewards, a significant loss of DAnergic neurons can result in movement disorders. Patients diagnosed with PD typically experience movement-related symptoms, including rigidity, slowness, and tremors [2]. In addition to motor symptoms, patients often suffer from non-motor symptoms, such as cognitive dysfunction and emotional disorders [3].

Because there is no permanent cure for PD and the symptoms progress without specific treatment, conventional treatments focus on slowing the neurodegeneration and managing related symptoms [2]. Conventional pharmacological treatments for PD primarily focus on alleviating motor symptoms by enhancing DAnergic neuron activity using levodopa, monoamine oxidase inhibitors, DA agonists, or combination therapies [2,4]. Currently, various alternative therapies for advanced PD are being actively explored, with several clinical studies comparing the efficacy and safety of novel options, such as surgical approaches and deep brain stimulation [5]. However, those treatments are symptomatic and do not reverse or halt the progression of PD, and sometimes they can be associated with side effects related to drug injections or invasive interventions, like headache, stroke, and depression [4,5]. Therefore, research to develop permanent treatments, including gene therapies and cell-based therapies, is generating increasing interest [5,6].

In various preclinical studies, researchers have sought to select the right PD animal model to better evaluate key factors, including construct, face, and predictive validity [7]. Since no animal model perfectly replicates PD clinical conditions, no single model can demonstrate validity across all criteria [8]. However, using an appropriated validated model improves the evaluation of the translational significance of data generated in preclinical studies [7,8]. Most PD gene therapy studies have utilized one of the following models: 1-methyl-4-phenyl-1,2,3,6-tetrahydropyridine (MPTP), a face validity model that replicates the PD phenotype in humans; 6-hydroxydopamine (6-OHDA), a model with both face and predictive validity, which can be used to predict currently unknown aspects of PD in humans; or synucleinopathy, a model with face and construct validity, in which the disease phenotype in animals reflects the currently understood etiology of PD in humans [7,8,9].

In this review, we examine gene therapy techniques for PD that aim to restore the functionality of the DAnergic pathway and prevent further neuronal degeneration by maintaining healthy DAnergic neurons. Viral vector-based gene therapies have been used to enhance DAnergic signaling [10,11] in PD patients. Moderate improvements in motor behavior have been reported and long-term follow-up is still being conducted (Table 1). Unlike symptomatic treatments, neurotrophic factor-based therapies are intended to restore and regenerate the DAnergic pathway [12,13]. Key candidates supporting the survival of DAnergic midbrain neurons are GDNF and neurturin [14]. Although postmortem analyses of patients treated with adeno-associated virus (AAV) serotype 2–neurturin therapy did not show improvement [15], ongoing studies continue to explore neurotrophic factors using various delivery strategies (Figure 1).

Factors in the midbrain that promote the survival or genesis of DAnergic neurons, such as Nurr1, are promising targets for this type of therapy [16]. In Nurr1-deficient mice, the maturation of DAnergic neurons results in progressive PD pathology, such as a reduction in DAnergic neurons and motor impairments [16,17,18]. Additionally, the ablation of Nurr1 in adult rodents leads to the reduced expression of genes associated with mitochondrial function and oxidative phosphorylation, suggesting that reduced Nurr1 expression might induce DAnergic neuron dysfunction, as well as the progression of degenerative changes [18,19]. In humans, Nurr1 expression is significantly diminished in the brains of both aged persons and sporadic PD patients [20,21], suggesting that it could be an promising target for PD gene therapy [16]. In MPTP mouse models, a face validity model that replicates the PD phenotype in humans [7,8,22], AAV(serotype 2)-mediated overexpression of Nurr1 and its co-transcription factor, Foxa2, in the midbrain, produced substantial neuronal recovery and caused a decrease in pro-inflammatory cytokines [22]. Tyrosine hydroxylase (TH)-positive DAnergic neurons were protected from toxic environments and striatal DAnergic neuronal fibers were preserved. Consequently, the restoration of motor behaviors associated with nigrostriatal DA transmission was observed through the cylinder, pole, beam, and locomotor tests, as well as the apomorphine-induced rotation test [22]. Mechanistically, the forced expression of Nurr1 and Foxa2 promoted the survival of DAnergic neurons and protected them against toxic factors [22]. For glial cells, Nurr1 and Foxa2 repress the transcription of pro-inflammatory cytokine genes mediated by nuclear factor kappa B, a central mediator in the inflammatory response, in a trans-repressive manner [23]. This approach might also be applicable to other neurodegenerative diseases, such as Alzheimer’s disease [24].

Gene therapy could modify a specific patient’s genes to replace, inactivate, or introduce a new or modified gene [14]. Gene therapies include gene silencing, gene replacement, and gene transfer, and often use viral vectors, including in PD clinical studies [10,25,26,27]. Gene therapy could prove to be a disease-modifying approach for PD, with the potential to improve the patient’s quality of life.

In this article, we mainly discuss clinical trials of gene therapies, primarily those using viral vectors, that target PD using different strategies and focus on molecules that have been thoroughly studied in recent years (Table 1). To incorporate relevant human clinical studies into our review, we performed a search of the clinicaltrials.gov database on 23 October 2024, using the keywords ‘Parkinson’ for condition/disease and ‘gene therapy’ for other terms. From the 77 results identified, we systematically excluded: (1) observational trials, or those with an intervention that was not gene therapy; and (2) studies not related to Parkinson’s disease patients. However, we did include observational studies that followed gene therapy trials. As a result, this selection process yielded 18 studies directly related to our focus area, which are discussed in detail in this review.

**Table 1 ijms-25-12369-t001:** Clinical studies on viral vector-derived gene therapies in Parkinson’s disease patients.

Completed, Terminated, and Ongoing Clinical Studies: Gene Therapy for Parkinson’s Disease					
Transferred Gene	Name of Study/Delivery Method	Phase	Duration	Result/Status	Reference
*AADC*	Intra-striatal infusion of AAV–hAADC-2	Phase 1	2004–2013	Improved UPDRS, including motor function, for 6 months.	Christine, 2009 [28] NCT00229736
	Intraputaminal infusion of VY-AADC01 (AAV–hAADC)	Phase 1	2013–2020	Improved motor function. Reduced medication requirement for 3 years.	Christine, 2019 [29] Christine, 2022 [30] NCT01973543
	Intraputaminal infusion of AAV–hAADC-2	Phase 1	2015–2018	No published results. Terminated.	NCT02418598
	Intraputaminal infusion of VY-AADC01 (PD-1102)	Phase 1	2017–2021	No published results. Completed.	NCT03065192
	VY-AADC02 (AAV2–hAADC) infusion into the brain (RESTORE-1)	Phase 2	2018–2024	Active.	NCT03562494
	Observational extension study for VY-AADC01 (PD-1101)	Observational	2018–2023	Improved motor responses compared to intravenous levodopa.	NCT03733496 [31,32]
*GAD*	Surgical infusion of AAV–GAD into subthalamic nuclei	Phase 1	2005–2008	Improved UPDRS score and PET signal after 3–12 months.	Kaplitt, 2007 [25] NCT00195143
	Bilateral surgical infusion of AAV–GAD into subthalamic nuclei	Phase 2	2008–2012	Improved UPDRS score for 6, 12 months.	LeWitt, 2011 [33], Niethammer, 2017 [34], Niethammer, 2018 [35] NCT00643890
	Long-term follow-up study of rAAV–GAD-treated subjects	Observational	2011–2012		
	AAV–GAD gene transfer into subthalamic nuclei	Phase 1 Phase 2	2022–2024	No published results. Completed in October 2024.	NCT05603312
	Long-term follow-up of GAD gene transfer in PD	Observational	2023–	No published results.	NCT05894343
*NRTN*	Intra-striatal delivery of AAV2–NRTN (CERE-120).	Phase 1	2005–2007	Improved UPDRS for 1 year.	Marks, 2008 [36] NCT00252850
		Phase 2	2006–2008	No benefit for 1 year.	Marks, 2010 [37] NCT00400634
	Bilateral intraputaminal and intranigral administration of CERE-120	Phase 1 Phase 2	2009–2017	No benefit for 2 years.	Bartus, 2013 [38] Warren Olanow, 2015 [39] NCT00985517
Fusion therapy (*TH*, *AADC*, *GCH1*)	Bilateral striatal injection of ProSavin^®^ (Lentivirus-TH, AADC, and GTP-CH1)	Phase 1 Phase 2	2008–2012	Improved UPDRS at 6 months, 1 year.	Palfi, 2014 [10] NCT00627588
	Long-term follow-up of patients who received ProSavin in a previous study	Observational	2011–2021	Improved ‘off’ UPDRS for 2 years. Terminated without further report.	Palfi, 2018 [11] NCT01856439
*GDNF*	Bilateral stereotactic convection-enhanced delivery of AB-1005	Phase 1	2012–2022	Safety was confirmed. Gene expression showed improvement. No further results.	Rocco, 2022 [40] Heiss, 2024 [41] NCT04167540
	Intraputaminal AAV2–GDNF delivery (REGENERATE-PD)	Phase 2	2024–	Recruiting.	NCT06285643
*GBA1*	Intra-cisterna magna injection of LY3884961(AAV–GBA1), 2 dose levels	Phase 1 Phase 2	2019–	Recruiting.	NCT04127578

## 2. Midbrain Dopamine Replacement Gene Therapy

Glutamic acid decarboxylase (GAD) catalyzes the conversion of glutamate into GABA throughout human synapses to regulate excitability. In PD, the GABA pathway, which inhibits hyperexcitability in the midbrain (including the substantia nigra), becomes impaired, leading to the dysfunction of nigrostriatal DAnergic neurons, the alteration of related brain circuitry and, eventually, movement disorder [42]. Downregulated *GAD1* gene expression has been observed in the neurons of PD patients [43]. Therefore, supplementing GABA through gene therapy was considered to have therapeutic potential. AAV–GAD treatment, which complements the enzyme responsible for GABA synthesis, was the first gene therapy to demonstrate effectiveness [25,44]. The first phase 1 clinical trial of the AAV–GAD treatment was conducted between 2005 and 2007 [25]. Another study showed evidence that the treatment modulated metabolic brain networks [45]. Those results led to a phase 2 randomized control trial that enrolled 44 participants with advanced PD and treated them with a bilateral infusion of AAV–GAD into their subthalamic nuclei. In follow-up studies from that research, AAV–GAD gene therapy produced significant motor improvements in PD patients and those improvements were maintained for more than a year [33,34] (Table 1). The changes included a unique functional connectivity alteration, indicating the formation of new connections between subthalamic nuclei and motor cortical regions, which was observed only in patients who received GAD gene therapy [35]. Currently, phase 1/2, placebo-controlled trials of a newly developed AAV2–GAD treatment delivered into subthalamic nuclei are ongoing to evaluate the safety and tolerability of two different doses of AAV2–GAD (NCT05603312) and a long-term follow-up study is planned for those patients (NCT05894343) (Table 1).

In the midbrain, L-tyrosine is converted into L-dihydroxyphenylalanine (L-DOPA) by tyrosine hydroxylase (TH), using the cofactor tetrahydrobiopterin-producing enzyme GTP cyclohydrolase 1 (GCH1). L-DOPA is further converted into DA by AADC. In patients with severe PD, enzyme activity is also decreased, causing defective dopamine synthesis. The depletion of AADC activity in PD patients has also been shown [46] and could be an important etiology of movement disorder. The importance of the AADC enzyme is further implicated in congenital depletion patients, because AADC deficiency produces movement-related disorders like PD. Therefore, AADC gene therapy could be a target for both PD and congenital disorders [47].

The overexpression of the *AADC* gene showed therapeutic potential in rodent and non-human primate PD model studies [48] (Table 1). Initial phase 1 clinical studies of an intraputaminal injection of AAV–hAADC showed that it was safe and had potential efficacy, with minimal side effects [26,28,49]. One trial included long-term follow-up [50,51] and showed the safety of the infusion, and PET uptake diagnosis of increased AADC expression implied constant expression of the transgene. To improve delivery, phase 1 trials using magnetic resonance imaging (MRI)-guided direct delivery of AAV2–hAADC were also conducted in a PD group (PD-1101 trial or VY-AADC01) [29] and it was shown to be a safe direct injection route for AADC therapy. In the VY-AADC01 trial, 3 years of follow-up indicated that the treatment was well tolerated, symptoms were stable or improved, and patients needed less medication [30]. Therefore, a randomized, placebo surgery controlled and double-blinded phase 2 clinical trial of a new AAV–hAADC therapy (VY-AADC02, NCT03562494) is ongoing, along with a long-term extension observational study (NCT03733496). Other clinical trials being conducted for AADC congenital deficiency are using gene therapy targeting AADC in the brain [52,53,54] to treat movement symptoms.

Another trial involving PD used a lentivirus as a vector to deliver combinational gene therapy. An open-label, dose-escalation phase 1/2 study treated participants with a lentiviral vector-based gene therapy targeting three genes: TH, AADC, and GCH1. This therapy, called ProSavin, demonstrated safety in the trial, although its efficacy was limited [10,11] (Table 1). The subsequent version of ProSavin, called AXO-Lenti-PD (formerly OXB-102, an improved gene expression cassette that is preclinically at least 5-fold more potent than ProSavin based on behavioral and movement analysis), was evaluated in a clinical trial completed in 2022. AXO-Lenti-PD was injected once with MRI-guided stereotactic delivery into the putamen and showed promising preclinical efficacy in non-human primates. However, the development of AXO-Lenti-PD was terminated (NCT03720418) when Sio Gene Therapies ceased the license agreement for the drug [55].

Additionally, a CRISPR gene activation system targeting TH was employed in an in vivo study of PD gene therapy. When sgRNAs targeting the rat TH promoter expressed in astrocytes were administered, dopamine production in the striatum of the 6-OHDA rat model (predictive and face validity model) recovered and motor asymmetry induced by the lesion was reduced [56]. This alternative approach could broaden the methods for applying effective gene therapies.

## 3. Midbrain Neurotrophic Factor Gene Therapy

Neurotrophic factors are essential for the survival, growth, differentiation, migration, and synaptic plasticity of neurons (Table 2). In PD, several neurotrophic factors are deficient, leading to the loss of DAnergic neurons. Therefore, neurotrophic factors have been considered potential targets for disease modification. Some clinical studies are focused on the direct infusion of those factors [57].

GDNF is a potent neurotrophic factor associated with survival and regeneration, as well as the maintenance, of DAnergic neurons. Since the initial discovery that GDNF can regenerate DAnergic neurons in vitro [58], several clinical trials have attempted to use a GDNF infusion to restore DAnergic neuronal systems [59,60,61]. Subsequent clinical trials have focused on intraputaminal injection of GDNF. Although those trials did not produce significant clinical improvements, they did demonstrate increased 18F-DOPA uptake [62]. Moreover, 18F-DOPA, as the tracer, has been used to demonstrate and quantify presynaptic DA function. Decreased 18F-FDOPA uptake has been reported in the striatum in PD patients and an increase in 18F-DOPA uptake indicates the functional recovery of DAnergic neurons. Further investigations have focused on MRI-guided intraputaminal delivery of AAV2–GDNF to enhance penetration and efficacy. Although that delivery–imaging combination system worked safely, it showed limited efficacy in human studies [40,62] (Table 1). However, a follow-up case study of an advanced PD patient enrolled in the same clinical trial reported that their PD symptoms were stabilized for 3 years [41]. Another ongoing phase 1b clinical study is now reporting that the treatment is meeting its primary target, with no serious adverse effects; AAV2-GDNF is well-tolerated in participants with PD, demonstrating general stability of the mild cohort. At 18 months, the moderate cohort demonstrated improvements in the Movement Disorder Society Unified Parkinson’s Disease Rating Scale (MDS-UPDRS), motor diary OFF time, unified dyskinesia rating scale (UDysRS) scores, and levodopa equivalent daily dose (LEDD) (NCT04167540) [63]. As a result, an ensuing phase 2 trial is ongoing (NCT06285643) and the indications are expected to be extended to the multiple system atrophy–parkinsonian type (NCT04680065).

Neurturin is a naturally occurring structural and functional analogue of GDNF that is encoded by the *NRTN* gene. Neurturin has been investigated for gene therapy applications, like those involving GDNF. Following promising results in animal studies [25,64,65], an open-label phase 1 clinical trial was conducted in 12 patients, who had been diagnosed with PD for at least 5 years. The patients were treated with bilateral, stereotactic, intraputaminal injections of AAV2–NRTN (CERE-120). Delivery was effective and no significant safety or tolerability issues developed in the patients [36] (Table 1). However, a subsequent phase 2 study to evaluate the safety and efficacy of AAV2–NRTN in a double-blind randomized trial did not demonstrate a pharmacological effect in patients with advanced PD after 12 months of intraputaminal injections [37]. A long-term and postmortem analysis of patients who received AAV2–NRTN therapy for 10 years revealed that although transgene expression could persist long term, the limited expression of neurturin did not produce clinical improvements [15,66].

The cerebral dopamine neurotrophic factor (CDNF) is another factor that could be used as a PD therapy. CDNF regulates endoplasmic reticulum (ER) stress-induced unfolded protein response (UPR) signaling and promotes protein homeostasis in the ER. Because ER stress can cause DAnergic degeneration in PD, CDNF could be a therapeutic target. In a 6-OHDA rat model, CDNF injections significantly reduced the degeneration of DAnergic neurons [67], and a brain-penetrating peptidomimetic compound based on human CDNF, called HER-096, also protected DAnergic neurons and reduced synuclein-aggregation and inflammation in a mouse synucleinopathy model of PD [68]. A clinical study of CDNF infusion, delivered using a drug delivery system with implanted catheters in each putamen, showed that the treatment was safe and well-tolerated in patients with moderately severe PD [57]. However, gene therapy using a viral vector was tested only in rodent 6-OHDA or paraquat-treated PD models [69,70] and an amyotrophic lateral sclerosis model [71] targeting motor neurons. These studies showed protection of the nigrostriatal pathway and motor recovery. Further clinical studies using CDNF to target PD motor dysfunction should be conducted to expand the drug’s target molecules.

The mesencephalic astrocyte-derived neurotrophic factor (MANF) is another neurotrophic factor for DAnergic neurons that regulates ER stress-induced UPR signaling and helps neuronal survival [72]. *MANF* gene delivery to a rodent 6-OHDA model showed efficacy in ameliorating motor deficits [73,74]. However, one study that used a combination of *CDNF* and *MANF* claimed that only the combined nigral overexpression of both factors led to behavioral recovery and dopamine neuronal protection [75].

The brain-derived neurotrophic factor (BDNF) is responsible for neurogenesis and an injection of BDNF could, thus, promote neurogenesis [76]. As a result, gene therapy trials of AAV2–BDNF injections in patients with neurodegenerative disorders, such as Alzheimer’s disease, are ongoing to supplement the trophic effect in remaining neurons in cognitive- and memory-related neural circuits (NCT05040217). In PD patients, BDNF expression in the substantia nigra is reduced [77]. AAV–BDNF therapy in mouse MPTP PD models attenuated motor and cognitive abnormalities [78] and those results could lead to further human trials.

The vascular endothelial growth factor (VEGF) is also a neurotrophic factor and could be a possible treatment choice [79]. Nanosphere encapsulated VEGF [80], AAV–VEGF injections [81], and VEGF-A expressing stem cells [82], were used in a 6-OHDA rodent PD model and neuroprotective effects, pain relief, and rotational behavioral recovery occurred after treatment. Also, preclinical therapeutic evidence supports the benefits of VEGF-B [83] and -C [84]. Further preclinical and clinical studies are warranted.

Neurotrophic factor gene therapy is attracting wide interest for its potential to produce neural regeneration. However, most clinical trials have shown low efficacy, which means that it cannot completely stop disease progression or regenerate brain function. Those findings are related to ineffective delivery, poor diffusion, and non-optimal doses of injected AAV. Recent results suggest that a combination of AAV–GDNF gene therapy with the cell transplantation of DAnergic neurons derived from human stem cells could be effective for enhancing survival and connectivity of transplanted midbrain dopamine neurons, leading to better anatomical integration and functional recovery, and effectively reconstructing DAnergic circuits [85]. Some kind of combination therapy could be needed to further develop neurotrophic factor gene therapy.

## 4. Gene Therapy Targeting Key PD Pathogenic Downstream Pathways

### 4.1. Targeting the Autophagy–Lysosome Pathway

Autophagy is an important mechanism for degrading and recycling neurotoxic molecules to maintain homeostasis and resist cell death. In PD patients, impaired autophagy could lead to pathologic accumulation of α-synuclein or other neurotoxins, which are risk factors for PD pathogenesis.

The *GBA1* gene encodes the enzyme lysosomal hydrolase glucocerebrosidase (GCase), whose deficiency causes Gaucher’s disease. Patients with Gaucher’s disease and heterozygous carriers of GBA mutations have significantly increased risk of developing PD, up to 5% at age 60 and 15–30% at age 80 [86]. Reduced GCase activity is also noted in PD patients, in whom it contributes to a vicious cycle of α-synuclein accumulation and GCase deficiency that produces lysosomal dysfunction [87,88]. GCase activity is also considered to be an important biomarker of the cognitive decline common in PD patients [89], suggesting that it plays a role in PD pathology.

Although enzyme replacement therapy is used clinically to treat patients with genetically inherited Gaucher’s disease, traditional therapies have been ineffective in addressing neurological symptoms due to their poor penetration of the blood–brain barrier [90]. Therefore, to relieve central nervous system (CNS) symptoms related to GBA1 gene abnormality in patients with Gaucher’s disease or PD, gene therapy that delivers GBA1 into the CNS could be an alternative approach. In animal models, intracerebral glucocerebrosidase gene therapy using the AAV vector in mice and macaques prevented and cleared α-synucleinopathy [91,92,93,94]. Based on those findings, an ongoing phase 1/2a, multicenter, open-label, ascending dose study, called PRV-PD101 (NCT04127578, PROPEL study), is evaluating the safety, tolerability, immunogenicity, biomarkers, and clinical effects of intracisternal high-dose and low-dose AAV9–GBA1 (LY3884961) administration in patients with moderate-to-severe PD who carry a GBA1 variant [95] (Table 1). Also, an open-label, phase 1/2, multicenter study, to evaluate the safety and efficacy of a single dose of LY3884961 in infants diagnosed with Type 2 Gaucher disease (NCT04411654), started in 2021 [96].

The transcription factor EB (TFEB) significantly regulates the coordinated lysosomal expression and regulation (CLEAR) gene network and promotes the autophagy–lysosome pathway, which can mitigate α-synuclein-induced toxicity [97]. In a GBA1-associated induced pluripotent stem cell model, TFEB expression was dysregulated, so it could contribute to GBA1-related pathology in PD. The overexpression or pharmacological enhancement of TFEB has shown potential in relieving α-synuclein toxicity and alleviating neuronal death, making it a viable disease-modifying therapy for PD [97,98].

### 4.2. Targeting Mitochondrial Activity with PGC-1α Gene Therapy

Mutations of the phosphatase and tensin homolog (PTEN)-induced putative kinase 1 (*PINK1*) and *Parkin* genes are the autosomal recessive findings most frequently diagnosed in PD. At the cell level, those genes control mitochondrial functions, including mitophagy-related quality control and mitochondrial biogenesis. Defects in those genes lead to an accumulation of the parkin interacting substrate (PARIS, ZNF746), which leads to the dysregulation of proliferator-activated receptor gamma coactivator-1α (PGC-1α) activation. Also, alpha-synuclein pathology could cause additional mitochondrial dysfunction that could lead to further cell degeneration [99,100,101]. Therefore, inducing mitochondrial biogenesis through PGC overexpression or downstream activation could be a therapeutic strategy for PD and other neurodegenerative pathologies [102]. In an MPTP mouse model that mimics DA neurotoxicity, delivering LV-PGC-1α into the striatum using a lentivirus corrected mitochondrial dysfunction [103].

### 4.3. Targeting ER Stress and the UPR: GRP78, XBP-1

When misfolded and aggregated α-synuclein accumulates in brain tissue, ER stress and the UPR are triggered inside cells, which causes neuron apoptosis [104,105]. Aggregated α-synuclein inhibits neurons from responding to misfolded proteins in the ER. A major ER chaperone, glucose-regulated protein 78 (GRP78), plays a key role in UPR regulation [106]. When GRP78 was overexpressed in rodent PD models, the UPR was regulated to inhibit apoptosis, which allowed more nigral DAnergic neurons to survive [107,108,109].

The X-box binding protein 1 (*XBP1*) gene is also important in the UPR-related IRE1/XBP signaling pathway, where it sustains brain functioning and delays the aging-associated phenotype [110]. Low XBP1 expression in the brain is linked with senescence and synaptic defects [110] and could be related to several neuropathies, including Charcot–Marie–Tooth disease [111]. In two different neurotoxin (MPTP and 6-OHDA)-based models of PD, AAV–XBP1 has shown neuroprotective effects in the midbrain and has reduced striatal denervation [112,113].

To summarize, targeting the autophagy–lysosome pathway, mitochondrial activity, and ER stress/UPR in PD is intended to mitigate disease progression, restore cellular homeostasis, reduce neurotoxin accumulation, and protect against neuronal death (Figure 2).

## 5. Potential Gene Therapy Targeting Transcription Factors in Midbrain Development

### 5.1. Nurr1 and Foxa2: Midbrain Factor Combination Therapy

Nurr1 (also known as nuclear receptor 4A2 [*NR4A2*]) is a master regulator of midbrain DAnergic neurons that interacts with other transcription factors [19,114]. It is crucial for the development of DAnergic neurons and regulates the transcription of AADC, TH, DAT, and VMAT2. Nurr1 is expressed early in embryogenesis and is important for maintaining both developing and mature adult DAnergic neurons [19]. Nurr1 expression is lost during aging and several degenerative processes, including PD [115,116]. Treatment with a Nurr1 agonist [117] or forcing the expression of the *NR4A2* gene could protect midbrain DAnergic neurons from degeneration [118,119]. Foxa2, a transcriptional factor that interacts with Nurr1, is essential for neuron survival, so it could also be a target for treating neurodegenerative processes. Gene delivery could target the nigrostriatal system, which degenerates in PD and causes primary motor symptoms.

In an MPTP PD mouse model, AAV-mediated overexpression of Nurr1 and its co-transcription factor, Foxa2, in the midbrain, rescued neurons and a considerable number of striatal fibers [22]. Additionally, researchers rescued neurons by forcing the expression of those factors in neighboring glia, which synergistically protected degenerating DAnergic neurons in a paracrine mode (Figure 3). Providing gene therapy in both neurons and glia promoted neurodevelopment and protected DAnergic neurons against various toxic factors by suppressing the expression of neurotoxic pro-inflammatory genes [22]. Astrocytes and microglia are resident glial cells in the brain. Although they typically have neuroprotective roles, their function is compromised in disease states, which can lead to harmful inflammatory conditions. The pathological transformation of glial cells, along with the degeneration of neurons, is a critical factor in the pathogenesis of PD and, thus, is a prime target for its treatment. During degeneration, the expression of Nurr1 and Foxa2 in DAnergic neurons declines [16,21] and glial cells shift toward a damaging reactive state. Neurons with diminished Nurr1/Foxa2 expression become susceptible to the toxic environment of the deteriorating midbrain (Figure 3, middle). Forcing the expression of Nurr1 and Foxa2 in DAnergic neurons supports their survival [22]. The expression of these transgenes in glial cells fosters a neuroprotective environment by reducing pro-inflammatory cytokine secretion and enhancing the release of neurotrophic factors (Figure 3, right).

It has been reported that GDNF failed to rescue DAnergic neuronal loss because of the toxicity of α-synuclein [120]. Because Nurr1 overexpression can protect DAnergic neurons from α-synuclein [120], combining AAV–Nurr1 with GDNF or NRTN in the midbrain might improve the outcomes of clinical trials and may prove to be a desirable alternative therapy for PD. Therefore, the co-expression of Nurr1 and Foxa2 for AAV-mediated gene delivery is a realistic therapeutic option [24].

### 5.2. Other Important Midbrain Transcriptional Factors

*LMX1A* gene therapy: In DAnergic neuron development, Nurr1, Lmx1a, and Pitx3 are representative transcriptional factors that determine the developmental fate of DAnergic neurons and the expression of those factors is downregulated in the DAnergic neurons of PD patients [120]. These genes can be targeted to inhibit DAnergic neuron degeneration, which is the main pathology of PD.

The LIM homeobox transcription factor 1A (*LMX1A*) and 1B (*LMX1B*) genes are needed for the maintenance and survival of adult midbrain DAnergic neurons. They control the expression of key genes involved in mitochondrial functioning and their ablation results in impaired respiratory chain activity, increased oxidative stress, impaired autophagy, and mitochondrial DNA damage. Therefore, these two genes are essential for the survival and maintenance of the brain [121].

In early-onset PD with a *DNAJC6* loss of function mutation, WNT-LMX1A signaling during DAnergic neuron development is dysregulated via impaired endocytosis [122]. This defect could be directly correctable through *DNAJC6* gene transfer, indicating that Lmx1a could be a key factor in PD pathology [123].

*PITX3* gene therapy: Paired-like homeodomain transcription factor 3 (or pituitary homeobox 3, *PITX3*) is a pivotal gene required for terminal differentiation during DAnergic neuron development and the maintenance of neuronal survival in the substantia nigra, through its interactions with other transcriptional factors and the WNT pathway [124]. *PITX3* polymorphisms could affect the risk of dementia and cognitive symptoms in PD [125].

*PITX3* gene therapy or cell therapy might be useful not only for helping DAnergic neurons to survive [126], but also for providing more neuroprotective effects to astrocytes. *PITX3*-transfected astrocytes could show neuroprotective effects through BDNF and GDNF expression [127].

## 6. Gene Editing and RNAi Therapies

Many methods of gene delivery, other than viral vectors, can be used for in vivo genetic modifications. Gene editing technology, such as CRISPR-Cas9, and base editing are promising therapeutic candidates. CRISPR/Cas9-assisted stem cell therapy to regenerate neurons or modify astrocytes to improve their DAnergic function are actively being studied [128,129,130] and direct base editing for neurons is also being tried in vivo [131].

RNA therapeutics are also possible choices for gene therapy. RNA-mediated interference (RNAi) is widely used in animal studies for therapeutic purposes. For example, the downregulation of alpha-synuclein (*SCNA*) has been studied by using shRNA, targeting *SCNA* expression [132,133]. Also, microRNA is emerging as a promising approach to regulating alpha-synuclein or modulating neuroinflammation [134].

Antisense oligonucleotides (ASOs) have garnered interest recently for targeting alpha-synuclein via the *SCNA* or *LRRK2* gene [135,136]. Mutations in leucine-rich repeat kinase 2 (*LRRK2*) cause lysosome dysfunction and are related to autosomal dominant familial PD. In LRRK-PD, ASOs are attracting interest as a promising method for gene therapy [137]. LRRK2 ASOs administered to mouse brains reduced LRRK2 protein levels and decreased the number of fibril-induced α-synuclein inclusions [136]. A phase 1 clinical study is currently evaluating the safety and pharmacokinetics of LRRK2 ASOs (BIIB094, REASON clinical study, NCT03976349) [138].

A combination of gene and cell therapy could also be considered. Cell therapy with DAnergic neurons in homotopic grafts resulted in poor innervation, but combining cell and gene GDNF therapy for the homotopic reconstruction of midbrain DA pathways facilitated the robust striatal innervation of homotopic DAnergic neuron grafts [85]. One recent animal study also tried chemogenetic neural circuit activation of striatal medium spiny neurons, using a retrograde AAV vector. That strategy for selective activation rescued defects in both mouse and primate models and could be used in humans to target circuit modulation [139].

## 7. Conclusions: Challenges and the Future of Gene Therapy

PD primarily affects small, specific areas of the brain, making it a suitable candidate for gene therapy. Targeted approaches allow for the precise delivery of therapeutic genes to affected regions, potentially restoring normal function or slowing disease progression. Currently, numerous gene therapy studies for PD have been conducted, and many more studies are underway, and they all aim to modify the course of the disease. However, most gene therapies are yet to be approved for clinical treatment of PD. Some failures have been due to inconsistency in terms of the treatment effects and reported adverse effects, and those are summarized in Table 3. Ongoing trials and future studies should be conducted based on this information.

The ideal goal of PD treatment is to ‘restore the DA pathway’. In addition to cell replacement therapy, only certain gene therapies involving midbrain factors are likely to achieve this. Gene therapy involving midbrain-specific developmental factors, such as Nurr1, Foxa2, and Pitx3, as well as neurotrophic factors like GDNF, BDNF, MANF, CDNF, and VEGF, holds promise for future therapy. With the approval of AAV vectors for gene therapy in Europe and the US [141], this approach will gain more traction in the coming years. Some serotypes of AAV are preferred as therapeutic vectors because of their favorable safety profiles, strong neural tropism, efficient and persistent induction of exogenous gene expression, and low immunogenicity in the brain, when compared with adenoviral and lentiviral vectors [142]. However, the risk-to-benefit ratio of treatments that involve permanent gene delivery must be carefully evaluated. Despite the risks, the lack of treatments that can stop the progression of PD indicates a significant unmet therapeutic need. In this review, we highlighted key insights from preclinical and clinical trials, including some that were unsuccessful but that may contribute to shaping the future of gene therapies for the treatment and management of CNS disorders. In conjunction with the current pharmacologically based clinical treatment paradigms, gene therapy could be used to address the needs of individual patients.

## Figures and Tables

**Figure 1 ijms-25-12369-f001:**
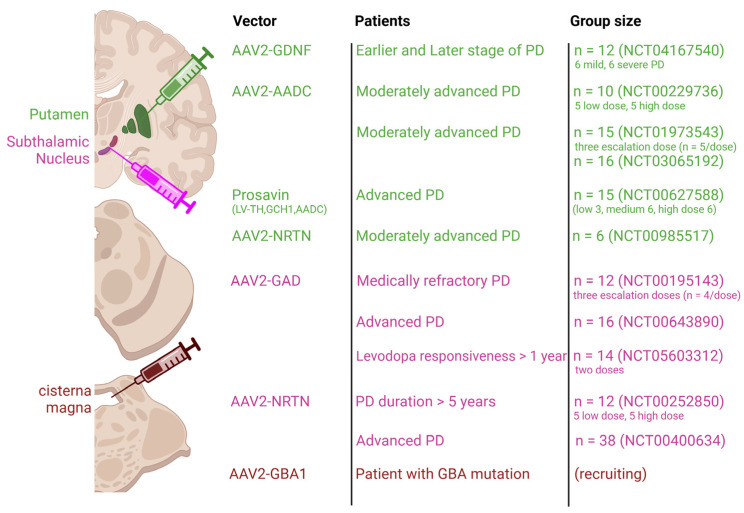
Representative gene therapy clinical trials for Parkinson’s disease. Injection route and expressed viral vector for gene therapy in Parkinson’s disease.

**Figure 2 ijms-25-12369-f002:**
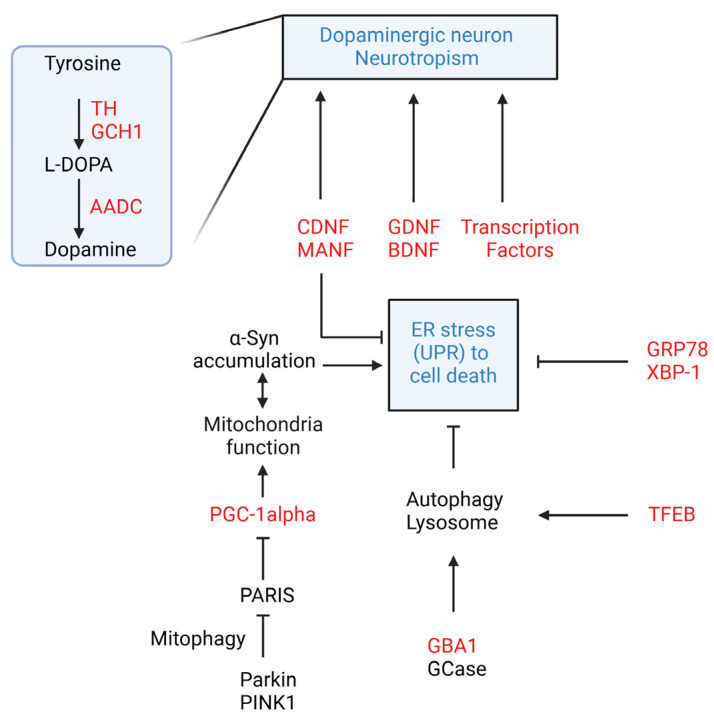
Major possible genetic targets of gene therapy in Parkinson’s disease and their related pathways. Targeting the accumulation of neurotoxins, such as alpha-synuclein, by modulating autophagy, enhancing mitochondrial biogenesis, or reducing ER stress-induced apoptosis could help preserve the dopaminergic cell population.

**Figure 3 ijms-25-12369-f003:**
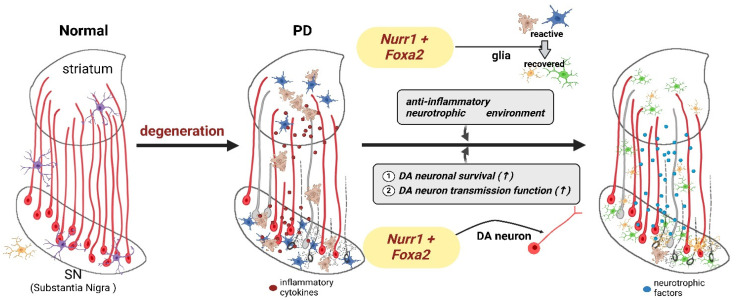
Schematic figure showing midbrain factor (Nurr1 + Foxa2) combination gene therapy. The nigrostriatal pathway links the substantia nigra (SN) pars compacta in the midbrain to the dorsal striatum (comprising the caudate nucleus and putamen) in the forebrain. In healthy dopaminergic (DAnergic) neurons, Nurr1 and Foxa2 are naturally expressed and contribute to cell survival autonomously (left, normal). The degeneration of DAnergic neurons in the SN impairs this pathway, leading to Parkinson’s disease (PD). With combination treatment in glial cells (microglia, astrocytes), the environment for neuronal survival is improved (↑). Also, midbrain transcriptional factors change the expression of healthy phenotypes in DAnergic neurons, including improved dopamine synthesis and neuronal signal transmission (↑).

**Table 2 ijms-25-12369-t002:** Midbrain neurotrophic factor genes related to PD pathology.

Abbreviation	Full Name
GDNF	Glial cell line-derived neurotrophic factor
NRTN	Neurturin
CDNF	Cerebral dopamine neurotrophic factor
MANF	Mesencephalic astrocyte-derived neurotrophic factor
BDNF	Brain-derived neurotrophic factor
VEGF	Vascular endothelial growth factor

**Table 3 ijms-25-12369-t003:** Limitations and adverse effects of previous PD gene therapy.

Transferred Gene	Name of Study	Limitations to Treatment Effects	Adverse Effects
*AADC*	Intra-striatal infusion of AAV–hAADC-2		In ten patients enrolled, three intracranial hemorrhages reported (one symptomatic, two asymptomatic cases)
	Intraputaminal infusion of VY-AADC01 (AAV–hAADC) (PD-1101)		One case of deep vein thrombosis with atrial fibrillation; four transient increases of dyskinesia
	Intraputaminal infusion of AAV–hAADC-2	No improvement in short-duration response to levodopa [140]	
*GAD*	Bilateral surgical infusion of AAV–GAD into subthalamic nuclei		One bowel obstruction, mild and moderate adverse effects (headache, nausea)
*NRTN*	Intra-striatal delivery of AAV2–NRTN (CERE-120)	Failed to show effect vs. sham surgery	
	Bilateral intraputaminal and intranigral administration of CERE-120	No significant difference vs. sham surgery	
ProSavin Fusion therapy (*TH*, *AADC*, *CH1*)	Bilateral striatal injection of ProSavin^®^ (Lentivirus-TH, AADC, and GTP-CH1)		Increased on-medication dyskinesias (20 events, 11 patients) and on–off phenomena (12 events, 9 patients). No serious adverse events
	Long-term follow-up of patients who received ProSavin in a previous study		Ninety-six adverse effects reported, including dyskinesia (11 patients, 33 events) and on-and-off phenomenon (11 patients, 22 events)

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
