# Peer review of "Gene Therapy for Parkinson’s Disease Using Midbrain Developmental Genes to Regulate Dopaminergic Neuronal Maintenance"

_ijms, 2024, doi:10.3390/ijms252212369_

Round 1
Reviewer 1 Report (New Reviewer)
Comments and Suggestions for Authors
The authors summarize approaches of gene therapy to treat Parkinson's disease (PD), which is significant to development of new therapy against PD. Although the review is comprehensive and in details, there are some concerns need to be addressed by the authors:
1, the gene therapy approaches have not yet achieved widespread clinical success, but several promising developments offer hope for future advancements. The authors have generated a table on gene therapy in PD. However, the final clinical outcomes are missing. The authors should add new content to the table about whether the therapy have improved the patient symptoms in short- and long-term observations.
2, The RNAi approaches is missing in the review. The authors only discussed the ASO. The authors should also include discussions on gene therapy of RNAi strategy.
3, The authors should add a section on "challenge and future" directions. The authors should summarize side effects of gene therapy in a new table. Obstacles and difficulty should be analyzed. Alternative approaches or combination approaches such as stem cell technology plus gene therapy should be discussed as well. The gene therapy to increase TH and AADC level may alleviate patient symptom transiently. However, PD is induced by progressive dopamine neuron loss. The enhanced dopamine secretion by remaining neurons can be deleterious, as dopamine is an endogenous toxin to dopamine neurons (Translational Neurodegeneration 12 (1), 44. 2023). The authors should discuss on it.
minor defect:
line 158: "was employed an in", should be "was employed in an in vivo study".
Author Response
Journal: IJMS (International Journal of Molecular Sciences)
Title: Gene Therapy for Parkinson’s Disease Using Midbrain Factors to Regulate Dopaminergic Neuronal Maintenance
Nov 11. 2024.
Dear, Reviewer
RE: Revision of the manuscript ijms-3268689(before)
Thank you for your careful evaluation of our paper and for the additional opportunity to revise it for publication in IJMS.
Based on your suggestions and comments, we revised our paper as follows.
Please find the revised manuscript enclosed together with our point-by-point responses to the comments made by the reviewer.
We are very appreciative of your support of our paper publication.
Mi-Yoon Chang, PhD
Department of Premedicine
College of Medicine, Hanyang University
133-791, Seoul, KOREA
Tel: 82-2-2220-0620
Fax: 82-2-2220-2422
E-mail: mychang@hanyang.ac.kr
Reviewer 1:
The authors summarize approaches of gene therapy to treat Parkinson's disease (PD), which is significant to development of new therapy against PD. Although the review is comprehensive and in details, there are some concerns need to be addressed by the authors:
1, the gene therapy approaches have not yet achieved widespread clinical success, but several promising developments offer hope for future advancements. The authors have generated a table on gene therapy in PD. However, the final clinical outcomes are missing. The authors should add new content to the table about whether the therapy have improved the patient symptoms in short- and long-term observations.
→ Thank you for your comment. As you suggested, we added a “Results/Status” section. The results/status section covers all short- and long-term follow-up results published. Including this, we updated many parts of Table 1, such as adding a new reference and moving it to follow the introduction.
2, The RNAi approaches is missing in the review. The authors only discussed the ASO. The authors should also include discussions on gene therapy of RNAi strategy.
→ Thank you for the suggestion. We changed the 6th Section to “6. Gene editing and RNAi therapies” and added details about RNAi therapies as follows:
RNA therapeutics are also possible choices for gene therapy. RNA-mediated inter-ference (RNAi) is widely used in animal studies for therapeutic purposes. For example, downregulation of alpha-synuclein (SCNA) has been studied by using shRNA targeting SCNA expression[132,133]. Also, microRNA is emerging as a promising approach to regulating alpha synuclein or modulating neuroinflammation[134].
3, The authors should add a section on "challenge and future" directions. The authors should summarize side effects of gene therapy in a new table. Obstacles and difficulty should be analyzed. Alternative approaches or combination approaches such as stem cell technology plus gene therapy should be discussed as well. The gene therapy to increase TH and AADC level may alleviate patient symptom transiently. However, PD is induced by progressive dopamine neuron loss. The enhanced dopamine secretion by remaining neurons can be deleterious, as dopamine is an endogenous toxin to dopamine neurons (Translational Neurodegeneration 12 (1), 44. 2023). The authors should discuss on it.
→ Thank you for this helpful suggestion. As our “Conclusion” section was mainly focused on the future of gene therapies, we changed the conclusion section to “Conclusion: challenges and the future of gene therapy in PD” and added relevant text. Also, we added Table 3, which addresses the limitations and side effects of PD gene therapies.
minor defect:
line 158: "was employed an in", should be "was employed in an in vivo study".
→ At the reviewer’s suggestion, we revised line 186 in the revised version of the manuscript.

Reviewer 2 Report (New Reviewer)
Comments and Suggestions for Authors
Gene therapy for Parkinson's disease (PD) is an evolving field aimed at correcting the underlying causes of the disease or providing neuroprotection by modifying the expression of specific genes. This review manuscript supplied a global viewpoint of the current status and progression of gene therapy in PD. There are several major and minor weaknesses in the rationale and research methods of this work. Below please find the review comments.
(1) Major comments
1. Rewrite the title. The author needs to revise the title to make it match the statement within this manuscript. For example, "Midbrain factors" is not a standard term in neuroscience or medical literature, it could be associated with proteins, transcription factors, or other molecular components that influence midbrain development or function in PD, so it is better to refer a more specific concept or scientific context.
2. The authors should update the key words after the abstract section. The key words should reflect the main topics, methods, results, and findings within this manuscript.
3. Please use the proper and correlated references to support the conclusion or statement. Ref. 2 mentioned both motor and non-motor disorders, it should be cited after the whole sentence. While Ref. 3 mainly talks about the Diagnosis and Treatment of PD, it is not suitable to support the non-motor deficient here. And all the following references need to be examined to meet this requirement.
4. Page 3, line 74, “suggesting that it could be an effective target for PD therapy”, I did not find the citation for this sentence, and I do not think this is a reasonable conclusion, it needs more data and evidence to support this point.
5. Page 3, line 97, I think it is required to add exclude and include criteria for the search results filter process. And the data access date is also required, as the database is updating all the time.
6. Page 3, line 109 needs references, similar issues happened through the introduction section, please double check.
7. Table 1 should support the conclusions in a scientific manner, it requires clarity, relevance, and alignment with the data/statement presented. It is difficult to get the conclusion in Page 3, line 115, please revise table 1 by filling in more details (such as the results/conclusion of each trial) to make it alignment with the main text.
8. Page 4, line 163, I think a table is required to summarize these midbrain neurotrophic factor genes.
9. Page 6, line250. Please double check when mentioning downstream midbrain target in PD, as these genes may not be located in the downstream of PD.
10. Page 9, line 354, please revise figure 3, I have no idea what it means “DA phenotype expression”.
(2) Minor comments
1. Abbreviations should be properly defined and interpreted when they are first introduced in the manuscript. Such as Parkinson’s disease and PD, the author uses these 2 terms randomly in this manuscript. This practice ensures that readers can understand the meaning of abbreviations without confusion. It is also recommended to include a comprehensive list of abbreviations after the conclusion section. This list will serve as a quick reference for readers, enabling them to easily access the definitions throughout the manuscript.
2. The authors should make the necessary modifications to the format of all the figures and tables to ensure they comply with the specific journal requirements, as some figures did not display well in my version, and the table format is not reader friendly. This may involve adjusting the table layout, font style and size, column headings, and other elements to align with the journal's guidelines.
3. There are several gramma issues, including missing/redundant markers or spaces. For example, Page 2, line 55 and line 62. missing/redundant spaces. Page 4, line 126, an incomplete sentence appeared.
Comments on the Quality of English LanguageNA
Author Response
Journal: IJMS (International Journal of Molecular Sciences)
Title: Gene Therapy for Parkinson’s Disease Using Midbrain Factors to Regulate Dopaminergic Neuronal Maintenance
Nov 11. 2024.
Dear, Reviewer
RE: Revision of the manuscript ijms-3268689(before)
Thank you for your careful evaluation of our paper and for the additional opportunity to revise it for publication in IJMS.
Based on your suggestions and comments, we revised our paper as follows.
Please find the revised manuscript enclosed together with our point-by-point responses to the comments made by the reviewer.
We are very appreciative of your support of our paper publication.
Mi-Yoon Chang, PhD
Department of Premedicine
College of Medicine, Hanyang University
133-791, Seoul, KOREA
Tel: 82-2-2220-0620
Fax: 82-2-2220-2422
E-mail: mychang@hanyang.ac.kr
Reviewer 2:
Gene therapy for Parkinson's disease (PD) is an evolving field aimed at correcting the underlying causes of the disease or providing neuroprotection by modifying the expression of specific genes. This review manuscript supplied a global viewpoint of the current status and progression of gene therapy in PD. There are several major and minor weaknesses in the rationale and research methods of this work. Below please find the review comments.
(1) Major comments
- Rewrite the title. The author needs to revise the title to make it match the statement within this manuscript. For example, "Midbrain factors" is not a standard term in neuroscience or medical literature, it could be associated with proteins, transcription factors, or other molecular components that influence midbrain development or function in PD, so it is better to refer a more specific concept or scientific context.
→ Thank you for these suggestions. We changed the title to “Gene Therapy for Parkinson’s Disease Using Midbrain Developmental Genes to Regulate Dopaminergic Neuronal Maintenance.”
- The authors should update the key words after the abstract section. The key words should reflect the main topics, methods, results, and findings within this manuscript.
→ Thank you for this helpful suggestion. We have revised the keywords to better reflect the main topics, results, and findings within this manuscript. Our revised keywords are: Gene therapy; Parkinson’s disease; Dopaminergic neuron; Adeno-associated virus; midbrain developmental genes; Nurr1; preclinical study; clinical trial; (Revised version Line 37)
- Please use the proper and correlated references to support the conclusion or statement. Ref. 2 mentioned both motor and non-motor disorders, it should be cited after the whole sentence. While Ref. 3 mainly talks about the Diagnosis and Treatment of PD, it is not suitable to support the non-motor deficient here. And all the following references need to be examined to meet this requirement.
→ We apologize for the confusion. We have revised the reference citations for Ref. 2 and Ref. 3 (revised version, Lines 47,48). Following the reviewer's comments, we also revised several other references.
- Page 3, line 74, “suggesting that it could be an effective target for PD therapy”, I did not find the citation for this sentence, and I do not think this is a reasonable conclusion, it needs more data and evidence to support this point.
→ Thank you for this helpful suggestion. We have changed the word 'effective' to 'promising' in the revised manuscript (Line 97). At the reviewer's suggestion, we have added more evidence to explain that the midbrain developmental factor Nurr1 could be a promising target (Lines 89-95.)
- Page 3, line 97, I think it is required to add exclude and include criteria for the search results filter process. And the data access date is also required, as the database is updating all the time.
→ Thank you for your suggestion. We made this section more detailed by including 1) exclusion and inclusion criteria, 2) (updated) data access date, and 3) additional details about the filtering process (Lines 116-125 of the revised manuscript).
- Page 3, line 109 needs references, similar issues happened through the introduction section, please double check.
→ Thank you for your comments. We have added the reference at line 133 (original version line 109), and we double-checked the introduction and revised the references.
- Table 1 should support the conclusions in a scientific manner, it requires clarity, relevance, and alignment with the data/statement presented. It is difficult to get the conclusion in Page 3, line 115, please revise table 1 by filling in more details (such as the results/conclusion of each trial) to make it alignment with the main text.
→ We agree with the reviewer’s comments and have revised Table 1 to include the results of each trial, and have moved it to follow the introduction (Line 127).
- Page 4, line 163, I think a table is required to summarize these midbrain neurotrophic factor genes.
→ In response to the reviewer’s comments, we have added information about neurotrophic midbrain factor genes in Table 2 (Line 193).
- Page 6, line250. Please double check when mentioning downstream midbrain target in PD, as these genes may not be located in the downstream of PD.
→ We apologize for the confusion. To clarify our meaning, we revised the subtitle to ‘Gene therapy targeting key PD pathogenic downstream pathways’ (Line 281).
- Page 9, line 354, please revise figure 3, I have no idea what it means “DA phenotype expression”.
→ We apologize for the confusion; the term ‘DA phenotype expression’ refers to ‘DA neuronal survival.’ In response to the reviewer’s comment, we revised Figure 3 to clarify this (revised Figure 3 & Lines 393-396).
(2) Minor comments
- Abbreviations should be properly defined and interpreted when they are first introduced in the manuscript. Such as Parkinson’s disease and PD, the author uses these 2 terms randomly in this manuscript. This practice ensures that readers can understand the meaning of abbreviations without confusion. It is also recommended to include a comprehensive list of abbreviations after the conclusion section. This list will serve as a quick reference for readers, enabling them to easily access the definitions throughout the manuscript.
→Thank you for your useful suggestion. For the readers' convenience, we have listed the abbreviations after the conclusion section and revised the abbreviations throughout the manuscript (revised version Lines 489-500).
- The authors should make the necessary modifications to the format of all the figures and tables to ensure they comply with the specific journal requirements, as some figures did not display well in my version, and the table format is not reader friendly. This may involve adjusting the table layout, font style and size, column headings, and other elements to align with the journal's guidelines.
→At the reviewer’s suggestion, we modified the tables and figures.
- There are several gramma issues, including missing/redundant markers or spaces. For example, Page 2, line 55 and line 62. missing/redundant spaces. Page 4, line 126, an incomplete sentence appeared.
→As advised by the reviewer, we had the revised version of the manuscript reviewed by an English-language editing service (eWorld Editing, Eugene, OR, USA).

Reviewer 3 Report (New Reviewer)
Comments and Suggestions for Authors
See attached file.

Author Response
Journal: IJMS (International Journal of Molecular Sciences)
Title: Gene Therapy for Parkinson’s Disease Using Midbrain Factors to Regulate Dopaminergic Neuronal Maintenance
Nov 11. 2024.
Dear, Reviewer
RE: Revision of the manuscript ijms-3268689(before)
Thank you for your careful evaluation of our paper and for the additional opportunity to revise it for publication in IJMS.
Based on your suggestions and comments, we revised our paper as follows.
Please find the revised manuscript enclosed together with our point-by-point responses to the comments made by the reviewer.
We are very appreciative of your support of our paper publication.
Mi-Yoon Chang, PhD
Department of Premedicine
College of Medicine, Hanyang University
133-791, Seoul, KOREA
Tel: 82-2-2220-0620
Fax: 82-2-2220-2422
E-mail: mychang@hanyang.ac.kr
Reviewer 3:
→ Thank you for your detailed comments. We have responded to each individually, as follows.
Abstract
(1) Lines 13-14: DA is used to abbreviate both “dopaminergic” and “dopamine.” One or the other should use the abbreviation.
→ We apologize for the confusion. To clarify the abbreviations, we revised the definitions to indicate that ‘DA’ stands for ‘dopamine’ and ‘DAnergic’ refers to ‘dopaminergic.’ Additionally, for the readers' convenience, we have listed the abbreviations after the conclusion section and revised them throughout the manuscript.
(2) Line 19: Indicate what the abbreviations AAV-GAD and AADC stand for.
→ We revised the abbreviation in Line 20 of the revised manuscript.
Introduction
(3) Lines 41-42: This sentience does not make sense. I needs to be rewritten.
→At the reviewer’s suggestion, we revised the sentence to: “Since DA regulates movement and the brain's response to rewards, a significant loss of DAnergic neurons can result in movement disorders.” (Line 44-45 in the revised manuscript)
(4) Line 47-48: Suggest you delete “for example pharmacological treatments”.
→At the reviewer’s suggestion, we revised the sentence.
We changed “Conventional treatment, for example pharmacological treatments…” to “Conventional pharmacological treatments” (Line 51 in the revised manuscript) to clarify and reduce redundancy.
(5) Line 55: Provide some examples of side effects.
→At the reviewer’s suggestion, we revised the sentences and added examples: “…related to drug injections or invasive interventions, like headache, stroke, and depression” in Line 58 of the revised manuscript.
(6) Line 60: What is meant by “activate gamma-aminobutyric acid (GABA)”? I do not think you mean activated GABA. It is not clear what is actually meant here. Do you perhaps mean activate GABA production?
→ In response to the reviewer’s comment, we deleted the sentences in question. The intended meaning was that ‘GABA therapies are aimed at modulating STN activity to improve the motor features of PD,’ which was mentioned in lines 101-122 of the original version.
(7) Figure 1: Need to provide references describing the different approaches illustrated in the figure and provide some detail on how the studies were done. Since it is stated that these are human clinical trials, it is important to include details such as AAV serotype, dosage, and disease stage at which the vectors were administered.
→ Thank you for the suggestion. As the reviewer said, we changed the Figure 1 to provide more information about AAV serotype, size and details about patient group as much as possible. Please find this new figure at Line 85 of the new manuscript.
(8) Lines 75-76: Explain what the MPTP mouse model is. How does this model what happens in human PD? Midbrain DA Replacement Gene Therapy Mave Table 1 up to this section closer to where it is cited in the text.
→ As the reviewer commented, we briefly explained the MPTP model (revised Lines 91-92). we added information about the MPTP and a frequently used rodent model (6-OHDA) in Lines 62-73 of the revised manuscript.
(9) Line 252: “Autophagy is the main mechanism for degrading and recycling neurotoxic molecules…” This is an overstatement. Replace “the main” with “an important” or similar wording.
→ At the reviewer’s suggestion, we revised Line 283 of the revised manuscript to “Autophagy is an important mechanism..”
(10) Line 287: What does PTEN stand for?
→ We added this information in Line 316 of the revised version: “Mutations of the phosphate and tensin homolog (PTEN)-induced putative kinase 1(PINK1)…”
(11) The authors refer to many different animal models for PD, but do not provide information on how well these models mimic human PD. Although the disease signs in the models may be similar to those of human PD, the underlying mechanisms may be different. It would be helpful to provide some information on how well validated the animal models are.
→ Thank you for your suggestions. Animal models have been useful in our understanding of the etiology of the disease and provide a means for testing new treatments. However, current animal models often fail to replicate the true pathophysiology of idiopathic PD, and thus the results from these models frequently do not translate to clinical outcomes. Although no model precisely recapitulates PD pathology, they still provide valuable information that contributes to our understanding of the disease and our treatment options. In response to the reviewer’s suggestion, we described the major PD animal models and outlined the types of validity for each model in Lines 62-73 (revised version).
(12) Lines 335-337: This sentence should be moved to the next paragraph which also includes information from the same study.
→ At the reviewer’s suggestion, we moved the sentence to the next paragraph (Line 368).
(13) Lines 366-367: This sentence should be moved to the end of the preceding paragraph.
→ At the reviewer’s suggestion, we moved the sentence to the previous paragraph (Line 401 of the revised manuscript).
(14) Table 1: The authors are inconsistent in providing the clinical trial reference numbers for the trials listed. Since it was stated that all of the trials included in this review were from the clinicaltrials.gov database, the trial accession numbers for all the trials should be provided for precision. This table need to be more appropriately placed in the paper.
→ Thank you for the comment. First, we added the trial accession number (NCT~) of every study in the ‘reference’ row of Table 1. Also, we moved the table to line 127, right after the introduction section where Table 1 is referenced.
(15) Lines 422-424: “The most ideal way to recover from PD is to 'restore the DA pathway.' Besides cell replacement therapy, only certain gene therapies involving midbrain factors can guarantee this effect.” These therapies do not “guarantee” the stated effect. Replace “guarantee” with “are likely to achieve”.
→ Following the reviewer’s helpful suggestion, we revised Line 472 of the revised manuscript to “…are likely to achieve this effect.”.
(16) Lines 428-430: “AAV is preferred as a therapeutic vector because of its favorable safety profile, strong neural tropism, efficient and persistent induction of exogenous gene expression, and low immunogenicity in the brain.” There are many AAV serotypes. The statement is not accurate for all serotypes. Much research is being done to optimize AAV serotypes for gene therapy applications.
→ As the reviewer suggested, we revised the text to “Some serotypes of AAV are preferred as therapeutic vectors because of their favorable safety profiles, strong neural tropism, efficient and persistent induction of exogenous gene expression, and low immunogenicity in the brain when compared with adenoviral and lentiviral vectors[142].” This provides more detailed and accurate statements about the AAV therapeutic vector (revised version Lines 477-480).
(17) Lines 433-434: “We highlight key insights from unsuccessful clinical trials …” Based on the descriptions in this paper, it is not accurate to imply that the clinical trials that were reviewed were uniformly “unsuccessful”
→ As reviewer noted, the conclusion was not entirely accurate. Therefore, we revised Line 483-485 of the revised manuscript to “we highlighted key insights from preclinical and clinical trials, including some that were unsuccessful but that may contribute to shaping the future of gene therapies for the treat-ment and management of CNS disorders..”

Round 2
Reviewer 1 Report (New Reviewer)
Comments and Suggestions for Authors
The authors have addressed all concerns carefully and revised their manuscript. The quality of the manuscript has been significantly improved. I suggest accepting their work in the current form.
Reviewer 2 Report (New Reviewer)
Comments and Suggestions for Authors
Thanks for your feedback, I am appreciating.
This manuscript is a resubmission of an earlier submission. The following is a list of the peer review reports and author responses from that submission.
Round 1
Reviewer 1 Report
Comments and Suggestions for Authors
This is a very interesting article in the context of Parkinson's disease, describing techniques that could also be used on other neurodegenerative diseases. It is well written and organized, which makes it easy to read and understand.
The bibliographic references seem very current and support what is written in the manuscript, the table Nº1 about clinical studies in this area is also an added value.
The only thing that I point out as most negative is figure 3, which should be revised. On the one hand it seems to have no definition and on the other hand what is presented in the PDF file does not seem to be the entire image, check this situation.
Point 1.: This review article aims to summarizes the available information about the gene therapies that have been developed over the recent years in an attempt to stop and reverse the Parkinsons disease progression. In my opinion, this is a very interesting and current topic, since the therapeutic approaches that exist are in an attempt to reduce the sequelae or symptoms and not to reverse the progression of the disease. Which in turn is also a topic that is transversal to other neurodegenerative diseases, such as Alzheimer's disease or ischemic stroke.
Point 2.: The article consists of an introduction that is well written and addresses the issue in general, highlighting the main challenges that the scientific community faces in the development of therapeutic approaches for Parkinson's disease, namely on genetic therapies. Subsequently, the article consists of 5 main topics organized by their putative order of importance. Starting with the genetic modulation associated with dopamine levels at the local level, a known crucial phenomenon in the development of Parkinson's disease. It then addresses the modulation of several neurotrophic factors (BDNF, GDNF, MANF, etc.) all associated with the improvement of neuronal function and in this case aimed at their effects on midbrain. Which is followed by one section that addresses other targets, which in this case are located downstream to the midbrain. Ending with two sections that look at the future, that on one hand addresses potential gene therapies and on the other hand to development of potential approaches. These 5 sections are well written, with clear language and the topics are relevant and well linked together, which makes it easier to read and understand the concepts explored.
Point 3.: The added value of the manuscript is the fact that it compiles into a single document the clinical trials carried out to date on human patients on this topic of research. This allows to quickly consult the available data, as well as their major conclusions, being this manuscript a starting point for consulting these studies.
Point 4.: As there is no section with the methodology for searching and selecting information, perhaps the authors should consider introducing a section explaining the way they searched and selected the information, the inclusion and exclusion criteria of the bibliography.
Point 5.: The conclusions are consistent with the evidence and the study’s that authors analyze in this review article. Highlighting that in the specific case of Parkinson's disease the future prospects are promising and in the coming years important steps will be taken not only in the treatment but also in the regression of the pathology itself. I personally believe that the main objective of the study was successfully achieved.
Point 6.: Yes, the bibliographic references are appropriate, well cited and well organized.
Point 7.: Regarding tables and figures I believe that table number one, which summarizes the clinical studies of viral vector–derived gene therapies in Parkinson’s disease patients it is very interesting and summarizes the studies that exist so far on this topic. The only thing that I point out as most negative is figure 3, which should be revised. On the one hand it seems to have no definition and on the other hand what is presented in the PDF file does not seem to be the entire image, check this situation
Author Response
"Please see the attachment."

Reviewer 2 Report
Comments and Suggestions for Authors
I thank the authors for the preparation of their manuscript, which gives an overview many recent studies developing or testing gene therapy approaches in efforts to treat Parkinson’s disease. The manuscript is quite comprehensive in covering the subject matter and give a fair balance over the various approaches. The quality of the review is, however, limited by the omission of some information needed to make a critical appraisal of the studies and to place their meaning beyond the headline outcome of each study. In many cases, the nature of the model used is unclear or the reasons underlaying the conclusions taken is unclear. In addition, certain statements are not sufficiently clear in their meaning.
I include a line-by-line commentary of these are other suggestions I make to the authors on how this could, in my view, be improved.
L20: It should be made more clear in the abstract what the distinction is between preserving dopaminergic neuron integrity and enhancing levels of dopamine the neurotransmitter.
L23: The authors say that ‘ … therapies targeting key downstream pathways, neurotrophic factors, and midbrain DA neuronal factors, and all of them have shown potential in pre-clinical and clinical trials.’. The authors should be more specific if they mean that there has been success in each of these categories overall, i.e. targeting key downstream pathways (though this is somewhat an ambiguous statement), targeting neurotrophic pathways, and midbrain DA neuronal factors.
L46: Authors state ‘conventional treatments focus on slowing or stopping the neurodegeneration’. A treatment which is effective at stopping neurodegeneration would surely qualify as a disease-modifying treatment, can the authors clarify what they mean by this statement.
L56: There is a lack of distinction between efforts to ‘restore DA pathway’ and efforts to maintain DA neurons themselves. Treatments such as deep brain stimulation do function by restoring DA pathway signaling (though via direct action on the DA neuron targets).
L73: The sentence ‘In PD models, AAV-mediated overexpression of Nurr1 and its co-transcription factor Foxa2 in the midbrain produced substantial neuronal recovery and a decrease in pro-inflammatory cytokines.’ does not include a clear citation. Additionally, the authors need to specify which PD models, which AAV and what exactly is meant by ‘substantial neuronal recover’ and whether this was associated with a functional recovery.
L96: The sentence ‘The first phase I clinical trial of AAV-GAD treatment was done between 2005 and 2007’, should have an associated citation.
L134: It is not clear in this section what the mechanism of the ProSavin lentiviral gene therapy is. The authors should describe what the viral construct actually contains, the basis for this and how the virus was delivered.
L140: It is unclear to me what is meant by ‘…directly revise a gene to produce specific molecules and activate pathway’, the authors should clarify this statement.
L141: The authors should use the plural of ‘astrocytes’, rather than the singular. Additionally, they should clarify the nature of the ‘th-gene edited’ astrocytes, as the description given is not sufficient to interpret results of the study as described.
L155: The authors here state that ‘intracerebroventricular injection of the GDNF gene were unsuccessful: the delivered gene did not reach the target tissues’. However, the cited study involves expression of GDNF protein and not GDNF gene.
L156: The authors should state the nature of the adverse effects associated with GDNF delivery. Were these adverse effects significantly associated with GDNF dosage, or were they rather incidentally observed effects.
L158: The authors state: ‘…they did demonstrate increased 18F-DOPA uptake’. Could the authors help the reader by explaining the meaning of the increase 18F-DOPA uptake and why this may indicate the possible efficacy of the treatment.
L162: For the citation given in ‘A case study of an advanced PD patient reported that PD symptoms were 162 stabilized for 3 years, and related gene expression was increased by a gene infusion[45]’, it should be clarified if this patient was enrolled on the previously mentioned studies or if this is independent of those.
L164: It should be specified what the positively improved primary endpoint of NCT04167540 was. Additionally, the authors should clarify that the citation refers to a non-peer reviewed outlet (conference proceedings).
L185: The authors should state what rat model was used in: ‘In a rat model, CDNF injections significantly reduced the degeneration of 185 DA neurons’.
L187: The authors should state what disease-modifying effects in ‘HER-096 also showed disease-modifying effects in a mouse model’
L190: The statement ‘A clinical study of CDNF infusions…suggests that increasing the permanent expression of dopamine in elderly brains could compensate for the pathologic expression of CDNF.’ is confused. The study boosts CDNF in brains potentially deficient in dopamine, rather than the other way round. Additionally, it should be clarify the patient group here, who are described only as ‘elderly’
L191: The authors should mention the outcomes and objectives of the studies of the ‘rodent PD models[54,55]’, as well as the models used.
L197: The authors should specify the rodent model in ‘MANF gene delivery to a rodent model showed efficacy in amelio-197 rating motor deficits’
L205: The authors should specify the mouse model used in ‘AAV-BDNF therapy in mouse PD models attenuated 205 motor and cognitive abnormalities, and those results could lead to further human trials’.
L210: The authors need further detail in the sentence ‘Different studies have shown that VEGF-A overexpression in rodent model studies could help ameliorate motor symptoms[65-67]’. What is VEGF-A, what rodent models, what motor symptoms?
L218: The authors mention: ‘Recent results suggest that a combination therapy of AAV-GDNF and cell transplantation could be useful for reconstructing the functional and anatomical circuits of midbrain DA neurons’ – but they need to explain what these recent results are and why we can take this meaning from them.
L281: The authors need to specify what the ‘neurotoxin-based models’ are exactly
L304: The authors need to specify what ‘PD mouse model’ Nurr1 and Foxa2a were overexpressed in.
L310: The authors need to specify what gene therapy was used in both neuronal and glia when mentioning the study under citation #15.
L323-328: The authors need to provide mentions and citations in the main body text for all of the statements in the legend of figure 3 about the mechanisms of Nurr1 and Foxa2a role in DA neuron survival.
L359: Some of the phrases in this paragraph are nonsensical. In the term ‘offering neuroprotection to astrocytes’, I believe the authors mean that PITX3 may provide neuroprotection via astrocytic expression, as astrocytes, by definition, cannot be ‘neuroprotected’.
L385: The sentence ‘The localization of substantia nigra pathology in PD has long been viewed as suitable for gene therapy.’ Is nonsensical. The authors should revise this.
Comments on the Quality of English LanguageThere are some indications of issues with written English, but these are covered in my specific comments about the lack of clarify or illogicality of certian statements. Generally English is of a reasonable quality and other than where it may have a role in these speicfic points it is sufficient.
Author Response
"Please see the attachment."

Round 2
Reviewer 2 Report
Comments and Suggestions for Authors
I thank the authors for the revised manuscript, in which they have made efforts to address my concerns and comments. Many of my comments have been adequately addressed, which I will not make specific mention of and instead focus on where I feel issues remain. While there are significant changes, there still remains some omission of important information about included studies and there is an overall lack of a critical perspective on the included studies or the synthesis of the findings of the review in the discussion. Additionally, there remain issues in the logic of the framing of different approaches and studies which inhibit the ability to form an overall view of the state of the field.
I include a line-by-line commentary of those issues which remain with the manuscript as I see them:
L20: The authors have made changes in response to me comment that they should make distinction between approaches to improving cell integrity of dopaminergic neurons and approaches which enhance dopamine neurotransmitter signaling or otherwise enhance network function. However, the distinction they have made does not seem fully logically. For instance the AAV-GAD and AADC gene therapies are described as ‘targeting DA neuron to improve cell integrity’, but the impact of AAV-GAD and AAV-AADC is to increase dopamine transmitter levels, which is therefore actually not impacting cell integrity but instead neurotransmitter signaling. I do not think therefore that the authors have understood this distinction and the manuscript has been made more rather than less confusing in its mechanistic distinctions.
L49 (prev. 46): The authors have adequately addressed my prior concerns here.
L59 (prev. 56): Similar to the abstract, the authors do not make a logical distinction between preservations or restoration of dopaminergic neurons themselves and approaches to boost dopaminergic signaling directly. Therefore, the statement here at the start, ‘In this review, we examine gene therapy techniques for PD that aim to restore the functionality of the DA pathway by maintaining healthy DA neurons and prevent further neuronal degeneration’ is inaccurate as they include gene therapy approaches which are not aimed to prevent neuronal degeneration or focused on DA neuron cellular health but rather are targeted directly at signaling networks. The inclusion of such papers is not a problem in itself but the inaccurate framing and contextualization presents an issue to usefulness of this as a review paper and would mislead more junior readers.
L77 (prev. 73): While the authors have added a significant amount of detail, there is not enough to fully understand the cited works. For instance, what is the species of the MPTP model, and what is actually meant by ‘neuronal recovery’? Such a term could cover several different effects of the AAV treatment and it is important to the readers that (in a few words) we know what kind of ‘recovery’ was seen here.
L90: The inclusion a systematic review element is interesting, but the authors could include some further details of the study conduct. For instance, typically one would include the search date that this was conducted and some additional information on the nature of how they interpreted their criteria for Parkinson and gene therapy. It in noticeable that the included list of 18 trials in Table 1 are all gene therapies with viral vector delivery methods and does not include for instance antisense oligonucleotide approaches such as BIIB094 (NCT03976349), but this is not mentioned as an inclusion criteria. Distinctions such as this are critical to understanding what is and is not covered by a review and to give confidence that the author are thorough within the scope of the paper. Additionally, the authors could clearly identify that Table 1 is instead these studies by citing it at this point.
L147 (prev. 134): The authors have added a good amount of detail needed to understand the ProSavin trial design. However, there still lack information of the mechanism of delivery, dosing regime, the distinction and rationale between the design of standard ProSavin and AXO-Lenti-PD. Furthermore, we do not know in what way the trial did not show efficacy. Was there indication for instance if the treatment was effective in increasing levels of TH, AADC and GCH1 (and therefore shows the delivery mechanism is effective), but was not effective in functional outcome measures? If it is not possible to determine this from the reported material then this should also be stated clearly, as such details can be expected to be given by the reader.
L177 (prev. 164): The authors still do not specify what is meant by ‘possible clinical benefit’ in this context and should offer some critical perspective on the weight of evidence that there is really clinical benefit, given that the use of ‘possible clinical benefit’ to describe it suggest it is not fully evidenced.
L202 (prev. 187): The authors still have not specified what the disease modifying effect is in relation to HER-096, and while they have specified that it is. A mouse model of synucleinopathy they have not specified the actual nature of this mouse model.
L203 (prev. 190): The authors still have not addressed the confused nature of this sentence in relation to CNTF infusions, no do they clarify the patient group of study. In addition the description of ‘using a drug delivery system’ cannot be interpreted by the reader and it is hard to understand what is actually being referred to here.
L206 (prev. 191): The authors make mention of a Paraquat-treated model but there is no mention of the outcome of this study.
L212 (prev. 197): The authors should specify the rodent (mouse or rat). How did the degree of rescue compare to CNTF or GDNF for instance? Did the combination of CDNF and MANF show benefits compared to CNTF or MANF alone? How might it differ to the action of the better-studied CNTF?
L334-340: The authors need to provide literature citations of all of the statements of biological fact within this section relating to figure 3.
L405-413 The amended section at the start of the conclusion section carries over much of the confusion between different approaches and how they relate as the prior sections. Some statements are made which are imprecise or unevidenced and overall there is little synthesis of the finding the authors have made in the main body of the paper. The authors should try to use the outcomes of the studies they have identified and what has been learnt from actually reading the papers to tell us something different to what could have been said in the introduction.
Comments on the Quality of English LanguageThere are some indications of issues with written English, but these are covered in my specific comments. Generally English is of a reasonable quality and other than where it may have a role in these speicfic points it is sufficient.
Author Response
Journal: IJMS (International Journal of Molecular Sciences)
Title: Gene Therapy for Parkinson’s Disease Using Midbrain Factors to Regulate Dopaminergic Neuronal Maintenance
Sep 9. 2023.
Dear, Reviewer
RE: Revision of the manuscript ijms-3144739
Thank you for your careful evaluation of our paper.
Based on the reviewer’s suggestions, we revised our paper as follows.
Please find the revised manuscript enclosed together with our point-by-point responses to the comments made by the reviewers.
The revised parts of the main text are marked in red (1st revision) and blue (2nd revision).
We are very appreciative of your support of our paper publication.
Mi-Yoon Chang, PhD
Department of Premedicine
College of Medicine, Hanyang University
133-791, Seoul, KOREA
Tel: 82-2-2220-0620
Fax: 82-2-2220-2422
E-mail: mychang@hanyang.ac.kr
Reviewer:
I thank the authors for the revised manuscript, in which they have made efforts to address my concerns and comments. Many of my comments have been adequately addressed, which I will not make specific mention of and instead focus on where I feel issues remain. While there are significant changes, there still remains some omission of important information about included studies and there is an overall lack of a critical perspective on the included studies or the synthesis of the findings of the review in the discussion. Additionally, there remain issues in the logic of the framing of different approaches and studies which inhibit the ability to form an overall view of the state of the field.
I include a line-by-line commentary of those issues which remain with the manuscript as I see them:
L20: The authors have made changes in response to me comment that they should make distinction between approaches to improving cell integrity of dopaminergic neurons and approaches which enhance dopamine neurotransmitter signaling or otherwise enhance network function. However, the distinction they have made does not seem fully logically. For instance the AAV-GAD and AADC gene therapies are described as ‘targeting DA neuron to improve cell integrity’, but the impact of AAV-GAD and AAV-AADC is to increase dopamine transmitter levels, which is therefore actually not impacting cell integrity but instead neurotransmitter signaling. I do not think therefore that the authors have understood this distinction and the manuscript has been made more rather than less confusing in its mechanistic distinctions.
→ Thanks for your comment, and we apologize for the confusion. Based on the reviewer’s comment, if we strictly distinguish between ‘improve cell integrity’ vs ‘increase DA transmitter level’, then the mode of action for AAV-GAD and AADC gene therapies would fall under ‘increase DA transmitter levels’. Reviewer’s criticism is absolutely correct, and to clarify the meaning, we revised the sentence in line 19.
L49 (prev. 46): The authors have adequately addressed my prior concerns here.
→ Thanks for reviewer’s feedback.
L59 (prev. 56): Similar to the abstract, the authors do not make a logical distinction between preservations or restoration of dopaminergic neurons themselves and approaches to boost dopaminergic signaling directly. Therefore, the statement here at the start, ‘In this review, we examine gene therapy techniques for PD that aim to restore the functionality of the DA pathway by maintaining healthy DA neurons and prevent further neuronal degeneration’ is inaccurate as they include gene therapy approaches which are not aimed to prevent neuronal degeneration or focused on DA neuron cellular health but rather are targeted directly at signaling networks. The inclusion of such papers is not a problem in itself but the inaccurate framing and contextualization presents an issue to usefulness of this as a review paper and would mislead more junior readers.
→ We understand the reviewer’s concern and moved 'by maintaining healthy DA neurons' to the end of the sentence to make its meaning clearer (line 58~59).
L77 (prev. 73): While the authors have added a significant amount of detail, there is not enough to fully understand the cited works. For instance, what is the species of the MPTP model, and what is actually meant by ‘neuronal recovery’? Such a term could cover several different effects of the AAV treatment and it is important to the readers that (in a few words) we know what kind of ‘recovery’ was seen here.
→ Thanks for your comment. We added the information in line 76 “mouse PD model”. As reviewer pointed, it’s important to describe a detailed description of neuronal recovery, so we have added the line 78~82.
L90: The inclusion a systematic review element is interesting, but the authors could include some further details of the study conduct. For instance, typically one would include the search date that this was conducted and some additional information on the nature of how they interpreted their criteria for Parkinson and gene therapy. It in noticeable that the included list of 18 trials in Table 1 are all gene therapies with viral vector delivery methods and does not include for instance antisense oligonucleotide approaches such as BIIB094 (NCT03976349), but this is not mentioned as an inclusion criteria. Distinctions such as this are critical to understanding what is and is not covered by a review and to give confidence that the author are thorough within the scope of the paper. Additionally, the authors could clearly identify that Table 1 is instead these studies by citing it at this point.
→ Thanks for your comment. Following the reviewer’s suggestion, to clarily the meaning of this review, we revised the sentence in line 92 to “ ..gene therapies, primarily using viral vectors, that target PD with ….”(Table.1). It seems necessary to move Table 1 forward to better align with the flow of the content.
L147 (prev. 134): The authors have added a good amount of detail needed to understand the ProSavin trial design. However, there still lack information of the mechanism of delivery, dosing regime, the distinction and rationale between the design of standard ProSavin and AXO-Lenti-PD. Furthermore, we do not know in what way the trial did not show efficacy. Was there indication for instance if the treatment was effective in increasing levels of TH, AADC and GCH1 (and therefore shows the delivery mechanism is effective), but was not effective in functional outcome measures? If it is not possible to determine this from the reported material then this should also be stated clearly, as such details can be expected to be given by the reader.
→ Following the reviewer’s suggestion, we added more detailed information in lines 150~154.
L177 (prev. 164): The authors still do not specify what is meant by ‘possible clinical benefit’ in this context and should offer some critical perspective on the weight of evidence that there is really clinical benefit, given that the use of ‘possible clinical benefit’ to describe it suggest it is not fully evidenced.
→ As reviewer’s comments, we added more detailed informations in line 185~188.
L202 (prev. 187): The authors still have not specified what the disease modifying effect is in relation to HER-096, and while they have specified that it is. A mouse model of synucleinopathy they have not specified the actual nature of this mouse model.
→ As reviewer’s suggestions, we added more detailed informations in line 210~211.
L203 (prev. 190): The authors still have not addressed the confused nature of this sentence in relation to CNTF infusions, no do they clarify the patient group of study. In addition the description of ‘using a drug delivery system’ cannot be interpreted by the reader and it is hard to understand what is actually being referred to here.
→ Based on reviewer’s comments, we clarified the CDNF treatment method and revised the information about the patient group in lines 213-216.
L206 (prev. 191): The authors make mention of a Paraquat-treated model but there is no mention of the outcome of this study.
→ As reviewer’s suggestions, we added the informations in line 218~219.
L212 (prev. 197): The authors should specify the rodent (mouse or rat). How did the degree of rescue compare to CNTF or GDNF for instance? Did the combination of CDNF and MANF show benefits compared to CNTF or MANF alone? How might it differ to the action of the better-studied CNTF?
→ Based on reviewer’s comments, we added the sentence in line 225~226.
L334-340: The authors need to provide literature citations of all of the statements of biological fact within this section relating to figure 3.
→ As reviewer’s suggestions, we added the citation in line 347 and 350.
L405-413 The amended section at the start of the conclusion section carries over much of the confusion between different approaches and how they relate as the prior sections. Some statements are made which are imprecise or unevidenced and overall there is little synthesis of the finding the authors have made in the main body of the paper. The authors should try to use the outcomes of the studies they have identified and what has been learnt from actually reading the papers to tell us something different to what could have been said in the introduction.
→ In response to the reviewer’s comments, to clarify the meaning of the review, we deleted lines 411~413 (original version). We also revised lines 423~4
Round 3
Reviewer 2 Report
Comments and Suggestions for Authors
Please see attached file

There are some indications of issues with written English, but these are covered in my specific comments. Generally English is of a reasonable quality and other than where it may have a role in these speicfic points it is sufficient.